Caregiver factors influencing family-based treatment for child and adolescent eating disorders: a systematic review and conceptual model

McCord Alex smccord@myune.edu.au 1 2
Rice Kylie 1
Rock Adam 1
1 School of Psychology, University of New England , Armidale , New South Wales , Australia
2 Child and Adolescent Specialist Eating Disorder Service, New South Wales Health , Northern NSW LHD , NSW , Australia
Cimino Silvia
Electronic publication date: 2025 Apr 9
Publication date: 2025
Volume: 13
Electronic Location ID: e19247
Received 2024 Oct 10; Accepted 2025 Mar 12
Copyright: ©2025 McCord et al.
Copyright year: 2025
Copyright holder: McCord et al.
License: This is an open access article distributed under the terms of the Creative Commons Attribution License, which permits unrestricted use, distribution, reproduction and adaptation in any medium and for any purpose provided that it is properly attributed. For attribution, the original author(s), title, publication source (PeerJ) and either DOI or URL of the article must be cited.
License URL: https://creativecommons.org/licenses/by/4.0/

Keywords: Adolescents, Eating disorders, Family-based treatment (FBT), Parental challenges, Caregivers, Anorexia nervosa, Bulimia nervosa, Family therapy

Funding: Australian Government Research Training Program Alex McCord is supported by an Australian Government Research Training Program (RTP) tuition fee offset. The funders had no role in study design, data collection and analysis, decision to publish, or preparation of the manuscript.

==============================
Introduction

The need to assess and manage familial factors influencing family-based treatment (FBT) has been identified in the literature in the context of improving outcomes. While some studies have attempted to address this need, results have not been unified into a framework and to date, no conceptual model exists to bring these factors together for use in clinical practice. A systematic review was conducted to fill this gap and addressed the following question: which caregiver factors influence FBT outcome for child and adolescent eating disorders?

Methodology

The protocol was registered in PROSPERO (CRD42022338843) and utilized the PRISMA framework. A total of 1,994 results were returned from EBSCO Host, Embase, ProQuest, PubMed Central, SCOPUS and Web of Science. Screening returned 164 studies for full-text-review with third-party replication to reduce risk of bias. Thirty-nine articles were included and organized in an evidence hierarchy including both quantitative and qualitative methodologies. Heterogeneity of the data precluded meta-analysis; results were synthesized and grouped using a systematic-narrative approach.

Results

Influential caregiver factors were identified and grouped into eight domains: caregiver capacity, confidence, readiness, internalizing factors, externalizing factors, food-related factors, support network and family function. Factors within each domain and their influence on treatment outcome were reported. A conceptual model, caregiver factors influencing treatment (Care-FIT) was produced as a graphical representation of the identified domains and factors by frequency of appearance.

Conclusion

Caregiver factors can significantly impact FBT outcome, and given the importance of their role in treatment, effective identification and management of caregiver factors is warranted. The conceptual model can be used in clinical case formulation and to support further exploration of the degree to which factors are influential. Identifying caregiver factors likely to influence treatment can facilitate support to enhance treatment and recovery.

Introduction

Family-based treatment (FBT) is currently a leading evidence-based treatment for adolescents with eating disorders (American Psychiatric Association, 2023; National Institute for Health and Care Excellence, 2017). In FBT, parents or custodial caregivers are seen as the key factor in overcoming the eating disorder (Treasure et al., 2021). Treatment is delivered in three phases, which include caregivers assuming control of renourishing their child in phase one. In phase two, meal autonomy is gradually returned to the young person, followed by phase three which addresses any other developmental concerns or difficulties along with relapse prevention planning (Lock & Le Grange, 2013; Rienecke & Le Grange, 2022). An essential component of FBT is active parental participation and leadership, specifically in the restoration of the young person’s weight to a healthy level by serving, supervising and encouraging completion of all meals (Adams & Hovel, 2024; Treasure et al., 2021).

FBT has been validated for anorexia nervosa (AN) and bulimia nervosa (BN; Forsberg & Lock, 2015; Gorrell & Le Grange, 2021) and its use is being evaluated with other child and adolescent eating disorders (Lock, Sadeh-Sharvit & L’Insalata, 2019; Van Wye et al., 2023). However, remission rates for AN in adolescents are about 40%, with the need to improve outcomes highlighted (Lock et al., 2024). An essential component of FBT is parental participation and leadership, specifically in the restoration of the young person’s weight to a healthy level by supervising and encouraging completion of all meals (Adams & Hovel, 2024; Treasure et al., 2021). Motivation to activate the required level of caregiver participation is grounded in psychoeducation about the negative impacts of malnutrition, and the framing of carer support as essential to recovery (Lock & Le Grange, 2013; Rienecke & Le Grange, 2022). It is inherently assumed that parents will be capable of shouldering both the leadership expectations and emotional distress; however, it has been suggested that parents experience challenges that may influence their implementation of treatment strategies, including emotions, cognitions, behaviors and practical obstacles (Giles et al., 2022; Marchetti & Sawrikar, 2024; Wufong, Rhodes & Conti, 2019).

Previous studies have focused on quantitative outcomes in research settings, community health and qualitative outcomes from parental and clinician perspectives (Astrachan-Fletcher et al., 2018; Cripps, Serpell & Pugh, 2024; Datta et al., 2023; Dimitropoulos et al., 2024). However, a wider reaching review including all these settings has not been produced. A scoping review of FBT trials examined non-specific predictors, mediators and moderators of treatment, finding several caregiver-related factors were relevant to outcome within the results including parental self-efficacy, expressed emotion, parental behaviors, treatment alliance and family conflict (Gorrell et al., 2022). Caregiver-level psychosocial factors such as blame, guilt or lack of empowerment, as well as accommodation of the eating disorder, family dysfunction, caregiver mental health or practical barriers have also been cited as negative influences on treatment delivery or outcome (Arikian et al., 2007; Keel & Haedt, 2008; Lavender, 2020; Loeb et al., 2012; Shimshoni, Omer & Lebowitz, 2021; Tasaka et al., 2017). The influence of cultural factors on family eating disorder treatment have been explored, such as traditional parental division of labor, culturally-sanctioned eating behaviors, collectivist or individualist values, with recommendations to adapt FBT or utilize cultural beliefs as a therapeutic resource (Chu, 2022; Dimitropoulos et al., 2024; Ram & Shelke, 2023; Williams, Wood & Plath, 2020). Parental strengths may include protective factors which may assist parents or caregivers in implementing FBT. Secure parent–child attachment, supportive family members, a willingness to learn about FBT, socio-economic stability or the ability to adjust commitments to support treatment have all been explored as potentially helpful (Basker et al., 2013; Jewell et al., 2021; Rienecke & Le Grange, 2022; Wallis et al., 2017b). These factors have thus far been assessed independently and have not yet been unified into a framework for evaluating strengths and challenges.

Several iterations of family treatment for eating disorders exist which emphasize the parental leadership role. In parent-focused treatment (PFT) the components of FBT are delivered separately with the clinician seeing the adolescent first, immediately followed by the caregiver session (Le Grange et al., 2016). PFT has been identified as potentially more effective than conjoined FBT in the presence of parental high expressed emotion (Allan et al., 2018; Eisler et al., 2000; Hughes et al., 2019). A number of intensive full-day programs incorporate FBT principles including intensive individual family therapy (IFT) delivered in approximately 40 h across one week by a multidisciplinary team, and multi-family therapy (MFT) in a group formats with five to eight families receiving four to five full days of treatment together across one week (Marzola et al., 2015; Rockwell et al., 2011). A longer-term day treatment program (DTP) is delivered in groups of five to eight families with 30 h of multidisciplinary treatment delivered weekly across approximately three months (Krishnamoorthy, Shin & Rees, 2023). It has also been argued that some families may not respond to individual FBT, and that the reduction in isolation and community created by a group setting in MFT as well as additional therapeutic activities may help support these families (Bruett et al., 2022; Dawson et al., 2018; Dennhag, Henje & Nilsson, 2021; Rienecke, 2017). MFT has also shown promising results when delivered as an adjunct to single-family FBT (Funderud et al., 2023). DTP has been shown to be effective and also support the FBT core tenet of parental involvement while supporting adolescents who require a higher level of care (Martin-Wagar, Holmes & Bhatnagar, 2019). Some differences exist between PFT, FBT and family therapy for anorexia nervosa (FT-AN); namely, PFT and FT-AN do not usually include a therapeutic family meal, and the four phases of FT-AN are condensed to three phases of FBT (Gorrell, Simic & LeGrange, 2023). However, the responsibility of the parent for nutritional rehabilitation is a key component of all derivations (Gorrell & Le Grange, 2021; Rienecke, 2017). While these versions of family treatment have demonstrated efficacy (Forsberg et al., 2023; Madden, Hay & Touyz, 2015), no protocol exists to assess caregiver factors, which could identify which format may be best suited to a particular family.

Gaps in the literature

Assessment of parents or caregivers for potential barriers at FBT commencement has been recommended (Accurso et al., 2021; Matthews et al., 2023). There is currently no consensus or protocol for caregiver assessment in the context of child and adolescent eating disorder treatment, for potential treatment barriers including factors which may interfere with the therapeutic process. It has been argued that further research to develop solutions and support for parents is needed in order to improve treatment outcomes (Giles et al., 2022; Gorrell & Le Grange, 2023; Van Huysse & Bilek, 2023; Wilksch, 2023). There is currently no protocol for the recommendation of a particular version of FBT to families. Given that FBT and its iterations consider the caregivers as crucial resources for their child in eating disorder treatment, it is important to consider parental capacity and barriers to participation in constructing a useful case formulation to provide treatment.

Research objective

The objective of this systematic review was to answer the following research question: which caregiver emotions, cognitions, behaviors, practical barriers or other factors relate to or influence FBT outcome for child and adolescent eating disorders? The review aimed to explore the challenges faced by parents or caregivers of a child or adolescent with an eating disorder, and how these challenges as well as any identified strengths or protective factors contribute to, or detract from, FBT outcome. A further aim of the research was to consolidate and analyze the findings within research and clinical practice settings including qualitative and quantitative methodologies. The information collected was synthesized and grouped to generate a conceptual model of caregiver factors likely to influence FBT treatment, for application in practice and research.

Methods

The research question was devised using the population, intervention, comparison and outcome (PICo) framework (Stern, Jordan & McArthur, 2014), and registered with the PROSPERO database of systematic reviews, ID number CRD42022338843. Preferred reporting items for systematic reviews and meta-analyses (PRISMA) guidelines were utilized for protocol development (Moher et al., 2009), with the research software Covidence used to screen database search results for duplication and relevance (Veritas Health Innovation, 2022). The nominated population included parents of children or adolescents with a diagnosed eating disorder listed in the Diagnostic and Statistical Manual of Mental Disorders, Fifth Edition, Text Revision (DSM-5-TR; American Psychiatric Association, 2022) for which FBT has been used. Included were: anorexia nervosa (AN), atypical anorexia nervosa (AAN), bulimia nervosa (BN), avoidant restrictive food-intake disorder (ARFID), binge-eating in the context of bulimia, and other specified or unspecified eating disorders (Forsberg & Lock, 2015; Gorrell et al., 2022; Lock, Sadeh-Sharvit & L’Insalata, 2019; Rienecke & Le Grange, 2022).

A Boolean search string was created by the research team (AM, KR, AR) and performed by AM/KR with terms derived from a scoping search of the literature using the terms eating disorders, parents and adolescents. A university librarian and a third-party research assistant reviewed the keywords and search strategy to address the risk of bias, and refine the search string. Keywords are listed by category in Table 1. Keyword categories included terms related to the involved adult/s, the intervention, the disorder, the client, and influencing factors. Databases searched included EBSCO Host, Embase, ProQuest, PubMed Central, SCOPUS and Web of Science. Searches were limited to articles published after and including the seminal literature on FBT published by Russell et al. (1987) until July 6, 2024. Results were restricted to peer-reviewed journal articles, theses and dissertations. Keywords were separated internally by the Boolean operator “OR” and between categories by the “AND” operator, with the asterisk “*” used as a wildcard to capture alternative spellings.

Table 1 Keyword search terms.

Adult	Intervention	Disorder	Client	
Parent*	Family-based therap*	Eating disorder*	Child*	
Famil*	FBT	Disordered eating	Adolescen*	
Caregiver*	F.B.T.	Anorexi*	Teen*	
Responsible adult*	Family therap*	Bulimi*	Young	
Principal care provider*	Maudsley method	ARFID	Youth	
Foster care*	Maudsley FBT	Avoidant restrictive	Youths	
Carer	Family-based treatment	Avoidant-restrictive		
Grandparent*	Family based treatment	Binge eating		
Grand-parent*		Binge-eating		
Influencing Factor	
Cognition*	Obstacle*	Hurdle*	Weakness*	
Cognitive	Rupture*	Barrier*	Influenc*	
Behavio*	Breakdown*	Difficult*	Unsuccessful*	
Emotion*	Hindrance*	Therapy interfer*	Success*	
Fail*	Complication*	Outcome*	Succeed*	
Challeng*	Impediment*	Strength*	Accommodat*	
Enabl*				
Notes.

Asterisks were included in the search string to capture alternative spellings.

Inclusion criteria reported in PROSPERO were (1) a client population of families with at least one adult caregiver of children or adolescents aged 6–18 with an eating disorder as defined by the DSM-5 TR (American Psychiatric Association, 2022). (2) The families were required to be engaged in a version of FBT including traditional conjoined FBT or FT-AN, PFT or FBT-informed treatments including DTP, IFT or MFT. (3) An assessment or interview of parents or caregivers must have been conducted. Exclusion criteria included (1) clients over age 18 or below age 6, (2) clients who did not live with adult caregivers or (3) where the caregivers were not assessed or interviewed, (4) the lack of an eating disorder diagnosis or diagnoses for which FBT is not typically used (e.g., pica or rumination), or (5) clients receiving another form of eating disorder treatment, such as systemic family therapy, acceptance and commitment therapy, cognitive behavioral therapy or dialectical behavior therapy, without adjunct FBT.

Preliminary searches returned a wide range of study methodologies, including randomized controlled trials (RCTs), quantitative and qualitative studies, studies reporting FBT clinician observations, and clinical case studies. While a wide variety of articles was intentionally sought in order to capture as many known influential factors as possible, it was identified that such broad results could limit the ability to draw conclusions. The decision was made to report results in an evidence hierarchy by study type according to health research standards (American Psychological Association, 2006; LoBiondo-Wood & Haber, 2022). Three study methodologies appeared in the search results which are classified as tier six, including qualitative studies with parents and with FBT clinicians, and clinical case studies. Consensus on inclusion of these methodologies was achieved based on the arguments that synthesis of qualitative and quantitative research is essential in health care research and should include all articles relevant to a research question (Dixon-Woods, Fitzpatrick & Roberts, 2001). It has been argued that eating disorder research on factors influencing outcome should include FBT clinician perspectives as they hold the responsibility for clinical decision making (Dimitropoulos et al., 2024). Consensus on the inclusion of clinical case studies was reached in accordance with the argument that psychological evidence-based practice should consider case study findings when change to assessment or process is proposed (Willemsen, 2022). In order to narrow the results, consensus was achieved among the authors to focus on eating disorders for which FBT has been validated in RCTs, resulting in the removal of studies focused on ARFID and binge-eating disorder.

From these revised criteria, 1,994 articles were returned including 1,970 from databases and 24 from citation searches. Citations were exported to EndNote and uploaded to Covidence. Duplicates were removed automatically with 691 studies screened by title and abstract against inclusion criteria by two independent parties; the lead investigator (AM) and a research assistant. Disagreements were resolved by the supervisory investigators (KR, AR), with 164 articles selected for full-text review. Initial quality assessment of articles was conducted by AM using the Joanna Briggs Institute critical appraisal tools for health care research specific to each study design (Lockwood, Munn & Porritt, 2015). A subsequent second round of quality assessment was completed by AM and replicated by an independent research assistant to ensure included articles had sufficient quality, academic rigor and a low risk of bias, using the McGill University mixed methods appraisal tool (MMAT; Hong et al., 2018) as studies were both qualitative and quantitative. Discrepancy resolution and confidence in the body of evidence were achieved through consultation with KR. This process resulted in 39 included studies which met MMAT quality and risk of bias standards, represented in Fig. 1. Studies were conducted in Australia (7), Canada (5), Europe (2), the United Kingdom (7) and the United States (18). The results of 3,782 measures or interviews were reported, completed by parents or caregivers to adolescent participants who were 86.9% female, with a mean age of 14.87 (0.74). Sixteen studies reported ethnicity, of which 77.95% of families were Caucasian. Twenty-nine studies reported family structure, of which 68.41% were intact families. Four studies reported observations of caregiver barriers from interviews with 117 FBT clinicians.

Figure 1 PRISMA flow diagram.

Heterogeneity of the data including varying study design, description of caregiver participants and analysis and variables measured excluded the possibility of meta-analysis; therefore, a systematic-narrative hybrid approach (Siddaway, Wood & Hedges, 2019; Turnbull, Chugh & Luck, 2023) was used and synthesis without meta-analysis (SWiM) guidelines were followed (Campbell et al., 2020). The Cochrane data extraction template (2023) was adapted to collect and organize the results by AM with oversight by KR/AR. Methodologies included 24 quantitative, 12 qualitative and three case studies. Twenty-four studies were classified tier 1, 2 or 3 including systematic reviews of RCTs, RCTs and quantitative studies, and fifteen studies were classified tier 6 including twelve qualitative studies and three clinical case studies. An overview and summary of included studies is organized by tier and first author, and presented in Table 2.

Table 2 Summary of included studies.

Evidence	Study	Location	Sample	Design	Eating disorder/s	Focus of study	Findings	Effect size reported?	
Tier 1	Datta et al. (2023)	USA	724 clients and their families	Analysis of Multiple RCTs	AN, BN	Impact of low socioeconomic status or non-intact family on FBT	A two-parent family and income level were not significant predictors of remission.	S	
Tier 2	Allan et al. (2018)	Australia	89 mothers, 64 fathers	Secondary RCT Analysis	AN	Impact of expressed emotion including criticism and emotional overinvolvement on treatment outcome	High maternal criticism was related to poorer outcome. Paternal expressed emotion was not significant to outcome.	No	
Tier 2	Baudinet et al. (2023)	England	167 families including 127 parents	Secondary RCT Analysis	AN, EDNOS	Moderators of treatment effect comparing single to multi-family treatment	Lower baseline positive caregiving experience was associated with lower weight at follow-up in individual family treatment. Perceived family conflict and negative caregiving experiences were not significant.	No	
Tier 2	Bohon et al. (2023)	USA	121 patients and their families, 95 intact and 26 non-intact, parental scores averaged	Secondary RCT Analysis	AN-R, AN-BP	Use of a lower threshold of expressed emotion to examine impact on early FBT or AFT response including criticism, hostility and emotional overinvolvement	Adolescents with one parent expressing even mild criticism, hostility or emotional overinvolvement were less likely to achieve early treatment response in FBT and AFT.	No	
Tier 2	Byrne et al. (2015)	USA	121 patients and their families, 95 intact and 26 non-intact, parental scores averaged	Secondary RCT Analysis	AN-R, AN-BP	Relationship of carer self-efficacy to weight gain	Increase in carer self-efficacy was related to weight gain.	No	
Tier 2	Darcy et al. (2013)	USA	21 patients, 16 intact families and 5 non-intact	Secondary RCT Analysis	AN-R, AN-BP	Impact of critical or negative behaviours on outcome	Fewer critical comments were associated with better outcome.	No	
Tier 2	Ellison et al. (2012)	Australia	59 clients; 83% intact families, 11% re-partnered, 6% single parent	Secondary RCT Analysis	AN	Components of FBT which predict weight gain	FBT dropout predicted by low parental control and maternal therapeutic alliance.	No	
Tier 2	Forsberg et al. (2014)	USA	38 families, 36 mothers and 25 fathers	Secondary RCT Analysis	AN	Relationship of the parent-therapist alliance to FBT outcome	Therapeutic alliance was similar between mothers and fathers, and did not predict recovery among treatment completers. Dropout sample insufficient for analysis.	M	
Tier 2	Forsberg et al. (2017)	USA	121 patients and their families, 117 mothers and 107 fathers	Secondary RCT Analysis	AN	Impact of parental mental health disorder on treatment outcome.	Higher weight gain was seen in FBT clients whose mothers had no/low depression. Parental symptoms improved post-treatment.	S-M	
Tier 2	Hughes et al. (2018)	Australia	198 clients, 194 mothers, 175 fathers, 165 siblings	Secondary RCT Analysis	AN, EDNOS	Impact of family attendance on separate and conjoined FBT outcome	Paternal attendance predicted outcome at end of treatment and was related to remission.	S-M	
Tier 2	Hughes et al. (2019)	Australia	106 families, 67 intact families, 39 non-intact	Secondary RCT Analysis	AN, EDNOS	Predictors of early response in separate and conjoined FBT	Lower criticism, father’s attendance and fathers’ therapeutic alliance were related to outcome	M-VL	
Tier 2	Le Grange et al. (2011)	USA	79 families, 78 mothers and 69 fathers	Secondary RCT Analysis	AN-R, AN-BP	Relationship of high expressed emotion to FBT outcome	Low expressed emotion was found in parents of participants. Maternal and paternal warmth were related to positive outcome.	No	
Tier 2	Lock et al. (2006)	USA	86 patients, 67 intact families, 19 non-intact (75% nuclear, 14% single parent, 11% reconstituted)	Secondary RCT Analysis	AN-R, AN-BP	Predictors of dropout and remission in FBT	Longer treatment predicted dropout. Reduction in problematic family behaviours predicted remission.	L	
Tier 2	Rienecke et al. (2016)	USA	121 patients, 106 mothers and 91 fathers	Secondary RCT Analysis	AN-R, AN-BP	Relation of carer expressed emotion and family function to FBT or AFT treatment outcome	Paternal criticism predicted poorer outcome. Maternal criticism was related to treatment dropout, and families with maternal hostility succeeded less in FBT. Other EE was not significant.	No	
Tier 2	Wallis et al. (2017a)	Australia	57 patients, 55 mothers and 46 fathers	Secondary RCT Analysis	AN-R, AN-BP	Attachment and family function in FBT as predictors of remission	Change in parent-reported family function did not predict remission. Higher levels of father behavioural control were positively related to remission. Adolescent-reported attachment did not predict remission.	No	
Tier 3	Dennhag, Henje & Nilsson (2021)	Sweden	24 clients, 23 mothers, 22 fathers	Quantitative/ Cohort	AN, EDNOS	Relationship of caregiver burden to treatment outcome	Higher maternal guilt pre-treatment predicted positive outcome. Paternal perceived burden of the child’s dysregulation negatively influenced outcome. Reduction in caregiver burden was associated with better outcome.	M-L guilt; S-M dysregulated behaviour and isolation	
Tier 3	Hagan et al. (2023)	USA	184 patients and their families - each parental score averaged	Quantitative/ Cohort	AN	Bridge pathways of carer self-efficacy in FBT	Weight restoration was predicted by parents’ belief in their responsibility to renourish their child.	No	
Tier 3	Jewell et al. (2021)	England	192 clients (129 intact, 24 single mother, 20 other)	Quantitative/ Cohort	AN-R, AN-BP, OSFED	Attachment and mentalization as predictors of treatment outcome in FT-AN	Excessive parental certainty about child’s mental state predicted poor outcome.	L	
Tier 3	Martin-Wagar, Holmes & Bhatnagar (2019)	USA	87 patients, 74 carers	Quantitative/ Cohort	AN-R, AN-BP	Relationship of carer empowerment to remission in DTP-FBT following medical instability or previous unsuccessful treatment.	A lower level of baseline parental empowerment predicted weight restoration.	L	
Tier 3	Matthews et al. (2018)	USA	51 patients, 51 parents (47 mothers and 4 fathers)	Quantitative	AN, AAN	Relationship of caregiver burden to perception of illness and potential FBT impact	The carers’ subjective experience of their child’s illness was more accurate in determining caregiver burden than objective measures.	M-L	
Tier 3	Matthews et al. (2023)	USA	114 patients, (85 mothers, 10 fathers, 2 other carers)	Quantitative/ Cohort	AN, AAN	Predictors of caregiver burden in FBT	Proactively evaluating caregiver burden before starting FBT is suggested. Caregiver predictors of burden were anxiety and family history of eating disorders.	L	
Tier 3	Nelson et al. (2023)	USA	47 families, 34 mothers, 5 fathers, 1 guardian and 7 unspecified caregivers	Quantitative/ Cohort	AN, AAN	Impact of caregiver eating behaviours on weight change during FBT	Caregiver intuitive eating was negatively associated with weight and shape concerns on adolescent symptomology, and positively correlated with weight gain.	No	
Tier 3	Robinson et al. (2013)	Canada	49 clients, 40 intact families and 9 non-intact with both parents participating	Quantitative/ Cohort	AN, BN, EDNOS	Relationship of self-efficacy to adolescents’ eating symptomatology in FBT, MFT and DTP-FBT	Increased empowerment led to improved carer self-efficacy, which predicted outcome. Early gains in fathers’ self-efficacy predicted reduction in eating disorder symptoms.	S-M	
Tier 3	Stewart et al. (2021)	England	180 patients, 35 parents	Quantitative	BN, EDNOS-BN	MFT-BN + Treatment as Usual (FBT or CBT-E) vs TAU only	Reduced parental depression post-MFT-BN, reduction in negative experiences of caregiving. MFT group less likely to drop out of outpatient treatment.	S-M	
Tier 6	Baumas et al. (2021)	France	3 clients, 4 mothers and 2 fathers	Qualitative	AN	Change in family dynamics following MFT	The burdens of time investment in treatment, scheduling and work were cited as barriers. Group cohesion was reported as an agent of change.	N/A	
Tier 6	Binkley & Koslofsky (2017)	USA	N = 1 with intact family	Case Study	BN	Providing culturally safe FBT	With appropriate understanding and cultural competency, FBT can be successful.	N/A	
Tier 6	Couturier et al. (2013)	Canada	40 clinicians	Qualitative	AN	Clinician perspective on factors influencing uptake of FBT	Influential factors included parental mental health, lack of motivation or awareness of illness severity, parental history of disordered eating, inability to attend sessions and desire for child to have individual therapy.	N/A	
Tier 6	Dimitropoulos et al. (2017)	Canada	30 clinicians	Qualitative	AN	Clinician perspective on parental empowerment in FBT	Empowerment influenced by parental mental health including history of an eating disorder, caregiver burden and burnout, inability to tolerate their own or their child’s distress.	N/A	
Tier 6	Egbert et al. (2024)	USA	9 clinicians	Qualitative	AN	Identifying clinician-perceived barriers to FBT delivery for families of low socioeconomic status	Barriers identified included caregiver lack of time and resources to adhere to FBT, food insecurity, psychiatric comorbidities and lack of family support.	N/A	
Tier 6	Joyce & Dring (2019)	England	N = 1 with intact family	Case Study	AN	Addressing relational containment during FBT	Unaddressed family grief or trauma can be an obstacle to FBT participation.	N/A	
Tier 6	Konstantellou et al. (2022)	England	17 parents	Qualitative	AN, EDNOS	Carer management of uncertainty in MFT	Intolerance of uncertainty reduced confidence and increased accommodation of illness, increases in exhaustion led to reduced ability to cope.	N/A	
Tier 6	McMahon, Stoddart & Harris (2022)	Scotland	15 fathers	Qualitative	AN	Contribution of fathers to FBT	Fathers highlighted the need for individual parent support and therapeutic alliance. They cited insufficient opportunities to openly discuss treatment challenges as a barrier to continued engagement.	N/A	
Tier 6	Murray et al. (2018)	USA	38 clinicians	Qualitative	AN	Mechanisms of therapeutic agency in FBT	Barriers to FBT included parental behaviours, expressed emotion, fear, lack of support and low competence.	N/A	
Tier 6	Peterson et al. (2016)	USA	N = 1 with non-intact family	Case Study	AN	Addressing parental emotion in FBT	Parental emotional dysregulation, lack of consistency between households and parental preoccupations with food influenced treatment.	No	
Tier 6	Socholotiuk & Young (2022)	Canada	5 parents	Qualitative	AN	Carer processes used during renourishment in FBT	Carer processes included creating the capacity to engage, negotiating care partnerships and goal setting.	N/A	
Tier 6	Thibault et al. (2023)	Canada	6 mothers	Qualitative	AN, BN	Maternal experience of FBT and family function	Negative influences on treatment included personal and family challenges, difficulties with the parenting role.	N/A	
Tier 6	Voriadaki et al. (2015)	England	6 mothers and 4 fathers	Qualitative	AN	Caregiver experiences of MFT	Parents reported that isolation could derail treatment.	N/A	
Tier 6	Williams, Wood & Plath (2020)	Australia	9 parents	Qualitative	AN	What is important to parents when participating in FBT	Parental therapeutic alliance influences outcome. It was important to clarify parents’ value and belief systems, providing extra support where necessary.	N/A	
Tier 6	Wufong, Rhodes & Conti (2019)	Australia	13 parents	Qualitative	AN	Carer experiences of FBT and MFT	Negative influences included low empowerment, negative emotion, lack of acceptance and poor parental alignment. If weight gain did not occur, parents reported self-blame and stigma of failure.	N/A	
Notes.

Tier 1: systematic reviews; Tier 2: randomised controlled trials (RCT); Tier 3: other quantitative studies; Tier 6: qualitative studies with parents or FBT clinicians, and clinical case studies.

AAN atypical anorexia nervosa

AN anorexia nervosa

AN-R anorexia nervosa, restrictive

AN-BP anorexia nervosa, binge-purge subtype

BN bulimia nervosa

EDNOS eating disorder not otherwise specified

AFT adolescent-focused therapy

Results

Synthesis of the data was achieved by grouping the findings as recommended by Campbell et al. (2020) into thematic domains. The initial factor groups were those identified during formulation of the research question: emotions, cognitions, behaviors or practical barriers. Within these groups, eight domains were identified following the systematic-narrative hybrid process (Siddaway, Wood & Hedges, 2019; Turnbull, Chugh & Luck, 2023). Specifically, each included article was organized by the independent variable or phenomenon of interest measured. As data was extracted, reoccurring variables measured within studies informed the generation of eight domains by AM with oversight from KR/AR. They are defined in the following section as: capacity, confidence, readiness, caregiver externalizing factors, caregiver internalizing factors, caregiver food-related factors, family function, and support networks. These domains were subsequently organized into a conceptual model using content analysis and frequency of appearance of each variable. It is represented visually in Fig. 2 at the conclusion of this section.

Figure 2 Care-FIT conceptual model of caregiver factors influencing treatment.

Note: Color graphic depicting a model of eight domains relevant to caregivers participating in eating disorder treatment, clockwise from top: externalizing factors, support network, food-related factors, family function, internalizing factors, readiness, confidence and capacity.

The review process highlighted that self-efficacy and expressed emotion, both well-known concepts or categories of behavior (Bandura, 1977; Brown, Birley & Wing, 1972), have been updated in the context of eating disorder research (Cherry et al., 2018; Rhodes et al., 2005). The definition of these concepts as well as methods of measurement varied within the results. This review attempted to synthesize the results by examining the individual qualities or behaviors attributed to expressed emotion and to self-efficacy, and reviewing the ways in which they influence eating disorder treatment.

Influential caregiver factors on FBT outcome

Capacity

Thirteen of the 39 articles returned were related to caregiver capacity. In the context of FBT, capacity has been defined as the personal, practical or emotional resources available to the caregiver to successfully participate in treatment (Socholotiuk & Young, 2022). Lack of resources was identified as contributing to poor FBT outcomes, including limited knowledge and understanding of the FBT model (Ellison et al., 2012), insufficient skill development and competence (Murray et al., 2018; Voriadaki et al., 2015), and limited time and practical resources necessary to devote to treatment such as organization of the family schedule to support meal supervision (Dimitropoulos et al., 2017; Egbert et al., 2024; Thibault et al., 2023). Higher perceived caregiver burden at baseline was associated with poor outcome measured by lower weight at follow-up (Baudinet et al., 2023). The level of burden on the primary caregiver was linked to lack of capacity and treatment dropout, including the need to care for other family members, or the inability to reorganize personal and professional schedules in order to provide meal supervision and support (Matthews et al., 2023; Thibault et al., 2023). Reduction in caregiver burden is associated with positive treatment outcomes (Dennhag, Henje & Nilsson, 2021). Additionally, poor physical or mental health of caregivers was identified as a factor driving lack of capacity and treatment dropout or unsuccessful outcome (Dimitropoulos et al., 2017; Lock et al., 2006; Matthews et al., 2023; Matthews et al., 2018; Murray et al., 2018; Stewart et al., 2021).

Confidence

A lack of confidence, empowerment or self-efficacy can negatively impact treatment, as revealed in 11 of the 39 articles returned. Confidence, appearing in 11 articles, and empowerment, appearing in seven articles, were defined similarly in the results as beliefs held by the caregiver that they can or cannot carry out FBT or support their child to recover (Darcy et al., 2013; Dimitropoulos et al., 2017). Ten articles defined self-efficacy as confidence, empowerment, or as related to the two qualities (Darcy et al., 2013; Datta et al., 2023; Dimitropoulos et al., 2017; Hagan et al., 2023; Martin-Wagar, Holmes & Bhatnagar, 2019; Matthews et al., 2023; Murray et al., 2018; Robinson et al., 2013). The significant at-home demands during FBT delivery require caregiver confidence, in particular to provide the meal support required to renourish the young person in phase one of FBT (Dimitropoulos et al., 2017). If carers are not confident in their ability to renourish their child, they may accommodate the eating disorder at mealtimes and not insist their child complete the meal (Konstantellou et al., 2022). If carers are worried whether FBT will be an appropriate therapy for their child, whether their child might need more individual therapy, or if they are caught in a preoccupation with the cause of the illness, carer confidence can be affected (Wufong, Rhodes & Conti, 2019).

A suggestion by Dimitropoulos et al. (2017) that carer confidence was related to empowerment was further investigated by Martin-Wagar, Holmes & Bhatnagar (2019), finding that carer empowerment predicted remission. Darcy et al. (2013) also suggested that empowerment is linked to confidence, and that together they are a vehicle by which self-efficacy is achieved. When carers begin to understand what is required to help their child achieve remission, both their confidence and empowerment increase, which in turn bolsters self-efficacy, which is predictive of outcome (Robinson et al., 2013). The results of this review indicate that higher carer self-efficacy predicted greater adolescent weight gain, with the reverse also supported that lower carer self-efficacy predicted lesser weight gain (Byrne et al., 2015). Weight restoration was predicted by lower caregiver empowerment at commencement of treatment (Martin-Wagar, Holmes & Bhatnagar, 2019). It was argued that clinicians should proactively identify which carer-level factors will build confidence, and that factors such as carer belief in their responsibility to weight restore their child can bolster self-efficacy (Hagan et al., 2023).

Readiness

Readiness in the context of this review was defined as a stage of change, whether the carer is ready to engage in treatment generally and FBT specifically (Robinson et al., 2013; Voriadaki et al., 2015). Seven articles returned related to readiness, which also related to acceptance, willingness and motivation. The methodology of Williams, Wood & Plath (2020) noted that whether participant families had received FBT or another form of eating disorder treatment had been determined in part by parental willingness. If carers do not accept it is their responsibility to renourish their child, they may lack readiness to engage in FBT and treatment is likely to stall (Hagan et al., 2023; Murray et al., 2018). The process of caregiver acceptance can also cause carers significant emotional distress if they are not ready to accept their child’s eating disorder (Thibault et al., 2023). Low parental or carer motivation, or a belief that professionals can help their child better than they can, also reduces readiness to engage and can negatively impact treatment (Couturier et al., 2013).

Caregiver externalizing factors

The inability of carers to regulate their behavior was identified as a factor which can negatively influence FBT outcome, returned in nine of the articles reviewed and including criticism, hostility, dysregulation and conflict avoidance (Allan et al., 2018; Rienecke et al., 2016; Murray et al., 2018). Criticism and hostility are qualities which fall within the larger concept of expressed emotion, studied in the context of eating disorder treatment as the behavior of a carer which is driven by the outward expression of their internal emotions related to the young person in their care (Szmukler et al., 1985). It has been acknowledged that the term expressed emotion is confusing, as it refers to external behaviors expressed as a result of the emotional relationship (Barrowclough & Hooley, 2003). This review discusses the behavioral components of expressed emotion individually within externalizing behaviors including criticism, hostility, and emotional overinvolvement as well as warmth and positive remarks (Le Grange et al., 2011).

Carer criticism of the young person is related to poor FBT outcome, including a longer duration of illness (Allan et al., 2018; Darcy et al., 2013; Hughes et al., 2019). Analysis of secondary RCT outcomes found that paternal and maternal criticism influenced treatment differently. Paternal criticism was found to predict less improvement in the young person’s disordered cognitions and behaviors, while maternal criticism was related to treatment dropout, and families with maternal hostility showed less improvement in FBT than a comparison group (Rienecke et al., 2016). Carer intrusiveness, exaggerated responses to the young person or over-protective behaviors fall under emotional overinvolvement (Leff & Vaughan, 1985), but were not shown in this review to predict poor outcome or dropout (Rienecke et al., 2016). Le Grange et al.’s (2011) study investigating the relationship between expressed emotion and FBT outcome did not find a relationship between carer criticism and FBT outcome, which they discussed could have been due to a small number of the sample expressing critical comments, and the suggestion that high expressed emotion may have led to non-participation in assessments resulting in lack of end-of-treatment data. However, a relationship was found between weight gain and carer warmth or positive comments (Le Grange et al., 2011). A further study built upon the limitations cited by Le Grange et al. (2011) to explore a lower threshold of expressed emotion in eating disorder treatment, finding that even low levels of expressed emotion could predict poor FBT outcome, with the recommendation to assess families at treatment commencement (Bohon et al., 2023).

The physical behavior of carers during meal support was also found to be related to outcome, with greater physical encouragement to eat (e.g., repeatedly presenting food during a meal, pushing or moving the child) associated with longer illness duration versus verbal prompting (Darcy et al., 2013). Supporting carers to remain calm when the young person is upset is highlighted in the FBT treatment protocol (Lock & Le Grange, 2013), and the present systematic review found that carer dysregulation was related to poorer outcome while the reverse, either less dysregulation or more positive behaviors such as warmth, was related to positive outcome (Allan et al., 2018; Darcy et al., 2013; Le Grange et al., 2011). Further, carer conflict avoidance such as giving up, negotiating with the eating disorder or not insisting the young person complete their meal due to concern over their ability to remain calm, was related to lower weight gain and poor treatment outcomes (Murray et al., 2018; Thibault et al., 2023). Reduction in problematic family behaviors predicted FBT remission (Lock et al., 2006).

Caregiver internalizing factors

Caregiver internalizing factors including emotion were found to relate to FBT in 12 reviewed articles. Emotion can influence whether carers are willing to accept FBT as a viable treatment (Couturier et al., 2013). Emotional distress about neglect of other family members resulting from emphasis on the young person’s eating disorder was cited as a negative influence in a qualitative study of parental experiences of FBT (Thibault et al., 2023). Fear for their child or for change in the family may also be a factor (Murray et al., 2018). Additionally, excessive carer certainty about their child’s mental state may predict poor outcome, which the authors suggest may be due to rigid thinking or difficulty accepting alternative perspectives (Jewell et al., 2021). The familial experience of grief and loss may overwhelm carers and shift focus away from the provision of meal support (Joyce & Dring, 2019).

Carers may experience reduced emotional resilience due to their own mental health conditions such as depression, anxiety or complex trauma, which may in turn reduce their ability to participate in FBT treatment (Dennhag, Henje & Nilsson, 2021; Egbert et al., 2024; Stewart et al., 2021). Caregiver anxiety can predict greater caregiver burden, which can negatively impact ability to engage in FBT (Matthews et al., 2023). Secondary measures in a treatment comparison RCT revealed that the incidence of parental psychological symptoms was low among participants and unrelated to outcome; however, maternal depression was a moderator of lower weight gain, and caregiver symptoms improved following FBT participation (Forsberg et al., 2017). In other studies where carers completed pre- and post-measures of psychological assessment, it was found that their symptoms improved as a result of participation in FBT and MFT (Stewart et al., 2021; Wufong, Rhodes & Conti, 2019). The relationship between caregiver burden and caregiver mental health is acknowledged, as well as the finding that a carer’s subjective experience is an accurate measure of burden with both potentially impacting FBT (Matthews et al., 2018). Higher maternal feelings of guilt pre-treatment predicted a positive outcome (Dennhag, Henje & Nilsson, 2021). Guilt experienced during treatment, either as a result of the FBT intense scene designed to activate parental motivation, or following from feelings of failure if their child lost weight, was cited by parents as demoralizing and a threat to treatment (Wufong, Rhodes & Conti, 2019).

Caregiver food related factors

Nine of the articles in this systematic review related to food-related caregiver factors. If caregivers have experienced an eating disorder themselves, caregiver burden may be increased and treatment for the adolescent may be at risk (Couturier et al., 2013; Dimitropoulos et al., 2017; Matthews et al., 2023; Peterson et al., 2016). Treatment barriers may stem from cultural beliefs which may require therapist sensitivity or adaptation of treatment delivery (Binkley & Koslofsky, 2017; Williams, Wood & Plath, 2020). Familial food or diet rules intended to support healthy eating may impact weight restoration (Hagan et al., 2023; Peterson et al., 2016). Additionally, food insecurity was highlighted as a potential driver of caregiver evaluation and choice of foods (Egbert et al., 2024). Examination of the relationship between caregiver eating behaviors and outcome including adolescent eating disorder symptoms and weight gain found that caregiver intuitive eating was positively correlated to weight gain, and negatively correlated to symptoms (Nelson et al., 2023). Peterson et al. (2016) also described the need to manage the familial history of eating disorders in a split household in order to support consistency between households to enforce renourishment within FBT.

Family function

Several factors within family function were found to directly impact treatment outcome, in 13 of the articles identified in this systematic review. The ability of all the young person’s carers or parents to participate in treatment, which could include biological parents and step-parents in the case of split families, is known as a factor influencing treatment outcome with particular focus on paternal attendance (Hughes et al., 2018; Hughes et al., 2019). Fathers participating in a qualitative study of the experience of FBT cited feeling excluded and insufficient opportunities to speak openly with clinicians as barriers to continued engagement (McMahon, Stoddart & Harris, 2022). A key component of FBT is carers and family members supporting the young person to eat by attending meals together and collectively providing support (Lock & Le Grange, 2013). If families have disparate schedules and are not able or willing to reorganize their commitments to engage in meal support, treatment is negatively affected (Couturier et al., 2013). Lack of parental alignment can impact treatment, for instance if meal support is not consistently carried out or the eating disorder is accommodated more by one carer or household, in the case of shared custody arrangements (Konstantellou et al., 2022; Peterson et al., 2016; Thibault et al., 2023).

The family ability to set boundaries around eating and limits on activity was related to successful outcome both for parents in general (Murray et al., 2018), and fathers specifically (Wallis et al., 2017a). Increased familial control of the young person’s eating can predict remission, and with that control can come increased conflict, which Lock et al. (2006) suggested may be an effect of carers successfully implementing the components of FBT. Conversely, low levels of parental control predicted FBT dropout (Ellison et al., 2012).

Perceived familial barriers such as low socio-economic status, marital quality or split families did not impede FBT, a finding supported at both RCT and individual case study levels (Datta et al., 2023; Peterson et al., 2016). The only included article which examined parent–child attachment as reported by the parent, found that attachment was predictive of therapeutic alliance, which predicted treatment outcome (Jewell et al., 2021).

Support network

Seventeen of the 39 articles in this review related the existence of support networks as influential on FBT outcome. The concept of negotiating necessary partnerships in order to renourish their child, whether with professionals, family or friends was a primary mechanism of change in the qualitative research Socholotiuk & Young (2022) conducted with families receiving FBT. Caregivers’ ability to emotionally support one another during treatment was also important, and when such support was lacking, primary carers were more likely to experience burnout (Thibault et al., 2023) or struggle to implement the treatment strategy (Egbert et al., 2024). Social isolation was a barrier to support from caregivers’ community due to stigma (Dimitropoulos et al., 2017; McMahon, Stoddart & Harris, 2022). Lack of connection to other carers of young people with eating disorders also contributed to parental distress, a factor addressed by group cohesion in MFT as an agent of change (Baumas et al., 2021). One case study found that without emotional support from their spouse about other familial issues outside their child’s eating disorder, the primary caregiver struggled to enforce meal completion (Joyce & Dring, 2019). Cultural sensitivity of the FBT therapist is also important. Some families may view the support or even participation of extended family, members of their church or other cultural or community networks as either necessary to support carers’ ability to complete FBT or contradictory to the organization of parental roles in the home (Binkley & Koslofsky, 2017; Williams, Wood & Plath, 2020).

The therapeutic alliance between clinician and parents is often strong in successful FBT, and was related to confidence in two reviewed studies (Hughes et al., 2019; McMahon, Stoddart & Harris, 2022). A stronger parent-therapist alliance predicted a positive outcome (Jewell et al., 2021), and the parent’s perception of support received from the therapist was empowering and instrumental in treatment success (Williams, Wood & Plath, 2020). Secondary RCT data also related the therapeutic bond to parental empowerment, but found that while therapeutic alliance was strong for both parents among treatment completers, it was not predictive of a successful outcome (Forsberg et al., 2014). A reported limitation of the study was a lack of sufficient dropouts for analysis in the sample. Hughes et al. (2019) found the clinician-father alliance predictive of early response, while Ellison et al. (2012) found that poor maternal therapeutic alliance predicted dropout.

Each reviewed study of MFT found that some carers believed they would not have been able to complete treatment without support, whether such resources were found in extended family, friends, their social network or other MFT participants (Baumas et al., 2021; Stewart et al., 2021; Voriadaki et al., 2015; Wufong, Rhodes & Conti, 2019). These findings complement research which found that improvement in carer perception of isolation was positively correlated to client remission in MFT (Dennhag, Henje & Nilsson, 2021), as well as the finding that parental unity in FBT predicted weight gain (Ellison et al., 2012).

Conceptual model

To assist clinicians and researchers in identifying factors influential to FBT, the following conceptual model of the review results was created systematically using content analysis and frequency of appearance of each variable. It is presented in Fig. 2.

Discussion

This systematic review included 39 studies which examined caregiver factors influential to FBT outcome including emotions, cognitions, behaviors, practical barriers and other factors. The recurring variables were measured by frequency of appearance, which resulted in the generation of eight domains including capacity, confidence, readiness, caregiver externalizing factors, caregiver internalizing factors, caregiver food-related factors, family function, and support networks. These domains and their constructs were organized into a visual conceptual model of caregiver factors influencing treatment (Care-FIT).

The review had the unanticipated result of highlighting the complexity of some well-known concepts within the field; that is, self-efficacy and expressed emotion. In the context of eating disorder treatment, Rhodes et al. (2005) defined self-efficacy as the ability of carers to adopt a primary role in managing the young person’s eating disorder treatment. The results of this review identified caregiver self-efficacy as predictive of successful FBT delivery and weight restoration (Byrne et al., 2015; Robinson et al., 2013). It was initially planned to include self-efficacy as a separate domain given the appearance of the concept in 18 of 39 articles; however, definition of the concept varied widely within the review results, with seven of the 18 articles not providing a definition. Eleven articles related self-efficacy to confidence or empowerment, or both, while four articles related self-efficacy to either ability, capacity, competence or resources. The present review explored these subcomponents individually and categorized them according to frequency of appearance. It is proposed that ability and competence are related to a caregiver’s capacity to carry out their part in the treatment, and that resourcing, caregiver health and caregiver burden affect one’s capacity to engage. The term carer empowerment was used interchangeably with confidence, and consistent with previous findings (Hagan et al., 2023; Robinson et al., 2013) it is suggested that both of these components can increase self-efficacy.

Criticism, hostility, emotional overinvolvement, warmth and positive remarks, were identified in this review as components of external expressed emotion. Bohon et al.’s (2023) findings grouped criticism, hostility and emotional overinvolvement together and suggested that the presence of any of these factors might negatively influence treatment. However, other articles in the results examined expressed emotion components individually. For clarity and to avoid the confusion noted by Barrowclough & Hooley (2003) surrounding whether these variables should be categorized as behaviors or emotions, the outward expression of each component was examined within caregiver externalizing factors.

Several caregiver externalizing factors identified in this review could potentially influence outcome positively or negatively. Le Grange et al. (2011) found that warmth and positive comments predicted weight gain. Further, the authors hypothesized that the absence of warmth might negatively influence outcome (Le Grange et al., 2011), which is in line with findings in this review that maternal criticism was related to poor outcome (Allan et al., 2018), and less criticism predicted better outcome (Darcy et al., 2013; Hughes et al., 2019). Likewise, factors contributing to caregiver burden could be positively or negatively related to outcome, in particular the carer’s perception of burden level, with high or increased burden negatively influencing treatment and feelings of empowerment, and lower or reduced burden leading to better outcomes as well as increased self-efficacy (Baumas et al., 2021; Couturier et al., 2013; Dennhag, Henje & Nilsson, 2021; Dimitropoulos et al., 2017; Matthews et al., 2023; Matthews et al., 2018; Stewart et al., 2021; Thibault et al., 2023). Parental control over their child’s activities appeared to potentially influence treatment positively or negatively, with low control predicting dropout and higher control related to remission (Ellison et al., 2012; Wallis et al., 2017a). It is suggested that if carer behavioral factors can be identified and managed, treatment success rates may improve, aligning with the findings of Lock et al. (2006).

Within internalizing factors, findings related to the influence of parental certainty on outcome appeared complementary. The finding that intolerance of uncertainty reduced confidence and increased parental accommodation of the eating disorder (Konstantellou et al., 2022) aligns with the finding that excessive parental certainty about their child’s mental state lead to poor outcome (Jewell et al., 2021). This systematic review results highlighted that guilt appears to affect treatment differently based on when it is experienced. Although the FBT model adopts an agnostic etiology of the disorder and attempts to minimize parental guilt (Rienecke & Le Grange, 2022), one included study found that higher pre-treatment maternal guilt resulted in a better outcome (Dennhag, Henje & Nilsson, 2021), while self-blame and feelings of failure experienced within treatment if weight gain did not progress, led to reduced empowerment and poorer outcome (Wufong, Rhodes & Conti, 2019). These findings support earlier conclusions that guilt arising if the young person did not achieve weight restoration in FBT phase one led to a poor outcome (Conti et al., 2017; Lavender, 2020), and perhaps suggests that clinicians should assess for pre-treatment guilt as well as monitoring parental emotion through treatment and offering additional support if weight restoration is challenging. The influence of parental mental health in general could potentially have a two-way impact, given that poor carer mental health increases burden and therefore poor outcome (Couturier et al., 2013; Dimitropoulos et al., 2017; Forsberg et al., 2017), while improvements in carer mental health and reduced burden were related to treatment completion and good outcome (Dennhag, Henje & Nilsson, 2021; Stewart et al., 2021). Finally, the finding that caregivers’ subjective perception of their child’s eating disorder treatment was predictive of outcome suggests that clinicians should be attuned to caregiver distress and offer additional support to minimize the risk of disengagement (Matthews et al., 2018).

Some elements of family function historically thought to impact treatment were found to be less conclusive in this systematic review. The finding that socioeconomic status or intact families do not predict remission (Datta et al., 2023) may help address historical concerns raised by clinicians that FBT is not appropriate for some families, and supports previous arguments that FBT can be successfully delivered to these families (Astrachan-Fletcher et al., 2018; Loeb & Le Grange, 2009). However, the reports of high burden, isolation and need for longer treatment among single parents (Doyle et al., 2009; Lock et al., 2005) should not be ignored in light of Datta et al.’s (2023) findings, particularly when considered with Lock et al.’s (2006) finding that longer treatment length was related to dropout. Attachment insecurity is known to negatively impact eating disorder treatment in adults and with different therapy modalities (Rossi et al., 2022); however, the results of this review linked attachment to outcome indirectly through therapeutic alliance (Jewell et al., 2021). This phenomenon was addressed by Wallis et al. (2017b), suggesting that the level of parent–child relational containment during FBT phase one nutritional rehabilitation may have a reparative effect on attachment, making an insecure or anxious attachment at commencement less of a risk to treatment outcome.

Intensive FBT-based programs studied in the present systematic review included DTP, IFT and MFT. These programs do not include any less carer involvement than traditional outpatient FBT; however, significantly more clinical time is spent with the young person and with the family in groups, which had the effect of increasing caregiver support (Girz et al., 2013). The finding that group cohesion in MFT was a mechanism of change (Baumas et al., 2021) supports a suggestion by parents that individual FBT may not be enough, particularly when caregivers or families are isolated. It may be helpful to expand the use of MFT as an adjunct to FBT to support increased remission rates, in line with Funderud et al.’s (2023) study that found significant improvement when families received concurrent MFT and FBT.

Limitations

The 39 articles included in this systematic review originated from the United States, Canada, Australia, the United Kingdom and Europe, which limits their generalizability to non-English speaking or non-western countries. While research returned in the database search from other countries (e.g., India, China and Japan), did not meet inclusion criteria for this review, they appear to align with the present systematic review’s findings that family function, cultural influences, and carer emotion and behavior (including acceptance of parental responsibility for renourishment) were all important factors to adolescent eating disorder outcome (Basker et al., 2013; Chu, 2022; Ram & Shelke, 2023; Sysko & Hildebrandt, 2011; Tasaka et al., 2017). Also, some included studies reported limitations that their participant samples consisted of treatment completers or that dropout factors were not analyzed (Forsberg et al., 2017; Forsberg et al., 2014; Le Grange et al., 2011), and thus there is a risk that not all treatment dropout factors were captured. Byrne et al. (2015) averaged caregiver Parents versus Anorexia scores rather than reporting results individually due to inconsistent parental attendance, which limits the ability to draw conclusions about differences between mothers and fathers or parental consistency within those families. Bohon et al. (2023) used a composite expressed emotion score combining criticism, hostility and emotional overinvolvement. Further, some studies (Allan et al., 2018; Hughes et al., 2018) noted the lower participation rates of fathers, which they suggested could have affected statistical power. Differences in healthcare systems between the United States and countries with universal or single-payer systems may account for differences in treatment completion. Access to treatment resulting from the healthcare system was cited as a barrier by Binkley & Koslofsky (2017).

Clinical implications and future research

The search that informed this systematic review was intentionally broad with the specific intention to bring together in one place all potentially influential caregiver factors. These findings, and the resulting conceptual model, have implications for clinicians working with FBT. Firstly, clinicians may routinely screen for caregiver strengths and barriers at assessment, which would allow them to tailor treatment with additions or modifications to FBT to support engagement. Secondly, the conceptual model presented in Fig. 2 of the current review could potentially be used as a tool to identify which form of FBT might be best suited to a family. For instance, it may follow that single-parent families could be offered MFT where available, or intensive PFT in addition to individual FBT. If treatment is occurring in an area where these derivations of FBT are not available, the conceptual model could assist the treating team to consider add-ons to therapy, or areas of focus which should be addressed in order to successfully complete treatment. These may include elements of CBT and DBT, which have been included in the intensive individual or MFT settings.

Further research is needed to develop and evaluate a protocol to assess caregiver factors and barriers which may influence treatment, and potentially identify strengths that may be utilized, so that families can be supported and collaboration between parents and clinicians can be strengthened. If a family profile indicates that a particular delivery method of FBT or a combined treatment modality may be more likely to succeed than others, the ability to provide tailored care based on individual families’ needs may reduce dropouts and increase remission rates. Future research may also test the conceptual model, potentially measuring the likelihood of a factor influencing treatment, or expanding or combining relevant domains.

Conclusion

Considerable evidence supports the RCT efficacy and real-world effectiveness of FBT and its permutations on child and adolescent eating disorder treatment, with guidelines recommending its use (Le Grange et al., 2021; Simic et al., 2022). While studies have explored individual factors which may increase or reduce effectiveness of FBT (Rienecke, 2017), this present study is the first to review, consolidate and organize these factors into a single conceptual model.

The present systematic review identified several caregiver factors which appear to positively or negatively influence the outcome of adolescent eating disorder treatment. These factors were grouped into a conceptual model including eight domains: caregiver capacity, confidence, readiness, internalizing factors, externalizing factors, food related issues, family function and the carer’s support networks.

It is hoped that this conceptual model will assist researchers in future experimental design to further explore the degree to which these factors influence treatment and whether some factors are bidirectional, or whether multiple factors are inter-related or have an exponential effect. The model may assist eating disorder clinicians with case formulation and treatment recommendations. Clinicians might use the model to consider whether adjuncts to FBT are appropriate, or whether a particular FBT format or version might be more suited to a family. Overall, it is hoped that assessment of caregiver factors at commencement of treatment could assist clinicians in recognizing possible treatment obstacles to be addressed and carer strengths which could support client recovery.

Supplemental Information

Supplemental Information 1 Rationale and unique contribution of this review

Supplemental Information 2 Prisma Checklist

Additional Information and Declarations

Competing Interests

Author Contributions

Data Availability

The authors declare there are no competing interests.

Alex McCord conceived and designed the experiments, performed the experiments, analyzed the data, prepared figures and/or tables, authored or reviewed drafts of the article, and approved the final draft.

Kylie Rice conceived and designed the experiments, analyzed the data, authored or reviewed drafts of the article, and approved the final draft.

Adam Rock analyzed the data, authored or reviewed drafts of the article, and approved the final draft.

The following information was supplied regarding data availability:

This is a systematic review/meta-analysis.

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
