# Peer review of "Caregiver factors influencing family-based treatment for child and adolescent eating disorders: a systematic review and conceptual model"

_PeerJ, doi:10.7717/peerj.19247_

## Round 0.1 · original submission · Major Revisions

Please pay particular attention to the comments from Reviewer 2

·

Basic reporting

Background, aims, methods and findings are presented in a clear and coherent way, and tables and figures are easy to read to support the manuscript very well.

I have only 3 small suggestions for further clarification:

a. in line 228 (results section) it is stated that the re-search question focused on “emotions, cognitions, behaviors and practical barriers”. This conceptual-ization is not clear from the Objectives section, which only states that focus is “caregiver factors”. I agree with authors that “emotions, cognitions, be-haviors and practical barriers” is a good operational-ization of “caregiver factors” – and I simply suggest that this operationalisation is made clear in the Ob-jectives section; this will also help set the context better for understanding the chosen search terms in the ”influencing factor”-box in table 1.

b. In line 231 it is stated that papers are organized by dependent variable. But is seems that papers are in fact organized by “caregiver factors”, which is in this context an INdependent variable, whereas out-come/remission is the dependent variable. If I have misunderstood, then I suggest adding some clarifi-cation for readers. Or else revise the wording of line 231.


c. Line 255 (the Results, Capacity section) mention ratings of caregiver “experience”, and it seems con-fusing in a section discussion caregiver capacity. Caregiver experience is a much broader (more vague) term than capacity, and it leaves the reader wondering what is meant here. Perhaps it is just an attempt to vary the wording, but it leads to more unclarity.

Experimental design

This manuscript has a clear methodology well suited for the research aim (however, with one reservation specified below), and the work adhere very well to recommended standards. Methods are elaborated and replicable.

The inductive organizing of papers into themes or care-giver factors is meaningful and well explained. For in-stance, the combination of terms such as confidence, self-efficacy and empowerment is s sound choice, and is well argumented with literature.

I have one consideration regarding the design; The Ob-jective state that they will examine how caregiver factors relate to as well delivery as outcome. Delivery and out-come are to rather different things, and may be too broad a scope, especially as the result section primarily revolves around the associations between carer factors and outcome. Also, the search string (and findings in Ta-ble 2) seems very well suited for answering the part about association between carer factors and outcome, but not so much the association between carer factors and delivery. Association to outcomes also receive the most attention in discussion, while only the last section is concerned with delivery.
I think the considerations on delivery in Discussions sec-tion are relevant, as delivery may impact the relationship between carer factors and outcome. But perhaps it should be kept as just a supplementary perspective in discussion (as in fact it appear now) but not be part of the objective, as it does not seem to be a central part if what this review can answer. As far as I can see, the ob-jective would be clearer and simpler and match the search better if delivery was omitted.
Or else, - if authors find that their review may answer the relationship between carer factors and delivery, then I recommend give it more attention in results section, to help the reader see these points.

Validity of the findings

Findings are well described, robust, and meaningfully placed within the FBT literature landscape.

Additional comments

Thank you for the opportunity to review this manuscript, which addresses an important knowledge gap. I especial-ly appreciate how authors bring all the evidence together in a clinically useful model (The Care-FIT). This is a sub-stantial contribution to the FBT literature.

Reviewer 2 ·

Basic reporting

This manuscript appears to have several issues, including numerous inconsistencies in abbreviation usage, which raise concerns about the level of care taken in its preparation. These issues include (non-exhaustive list based on reading the manuscript up to line 165, out of a total of about 600 lines in the manuscript main body):

- The Abstract contains abbreviations/acronyms such as "PICo", "SWIM" and "Care-FIT", which are undefined and uncommon, potentially causing confusion for readers.

- In the Abstract, the introduction states: "A systematic review was conducted to fill this gap and addressed the following question: Which caregiver factors influence family-based treatment (FBT) delivery or outcome for child and adolescent eating disorders?". However, in the conclusion, it is written: "The Care-FIT model can be used in clinical case formulation and to support further exploration of the degree to which factors are influential. Identifying caregiver factors likely to influence treatment can facilitate support to enhance treatment and recovery". This presents an apparent disconnection between the question posed in the introduction and the focus of the conclusion. While the Abstract begins by framing the systematic review as seeking to identify caregiver factors influencing FBT delivery or outcomes, the conclusion instead highlights the utility of the Care-FIT model for clinical application and future research. The lack of a clear link between the posed question, the findings, and the model's role in addressing the question creates a sense of inconsistency. Strengthening the connection between the study’s initial question, its results, and the Care-FIT model would improve the Abstract's coherence and alignment.

- On line 53, the term "family-based therapy (FBT)" is introduced without a clear definition. This lack of definition could pose a significant challenge for readers, especially since this concept appears to be a central theme of the manuscript.

- Line 69, it is written: "Previous studies have focused on quantitative outcomes in research settings, community health and qualitative outcomes from parental and clinician perspectives". However, this statement is presented without any supporting references, which may lead readers to question its validity and the evidence underlying this claim.

- The paragraph starting line 90 contains several terms that may be confusing to readers due to a lack of clear definitions. These terms include "parent-focused therapy (PFT)", "individual family intensive (IFT)", "multi-family therapy (MFT)", and "day treatment program (DTP)". One specific issue concerns the term "individual family intensive", as it appears to be missing a qualifier such as "therapy", "treatment", or "program". Additionally, it is incorrectly abbreviated as "IFT" rather than "IFI", in accordance with standard abbreviation rules.

MINOR POINTS:

- Line 58, is it written: "adolescent AN (Anorexia Nervosa)". It may be more accurate to say "AN in adolescents", which specifies the population of interest without implying a distinct subtype.

- On line 126, the abbreviation "FBT" for "family-based treatment" is defined again, despite being previously defined on line 53 and used multiple times between lines 53 and 126.

- The term "PROSPERO" is written inconsistently, appearing in uppercase on line 136 but in lowercase on line 160.

- On line 163, the abbreviation "FT-AN" for "family therapy for anorexia nervosa" is defined again, despite being previously defined on line 105.

- On line 164, the abbreviation "PFT" is defined as "parent-focused family-based treatment", whereas on line 91, the same abbreviation was defined as "parent-focused therapy". This is an issue because repeating the definition of an abbreviation is unnecessary and goes against standard writing conventions, which state that an abbreviation should be defined only once when first introduced. Furthermore, providing inconsistent definitions for the same abbreviation, as seen here with "PFT," creates confusion for readers and undermines the clarity of the manuscript. Such inconsistencies and redundancies suggest a lack of attention to detail and may negatively impact the reader's ability to follow the text effectively.

Experimental design

Please see the "Basic reporting" section

Validity of the findings

Please see the "Basic reporting" section

Additional comments

Please see the "Basic reporting" section

·

Basic reporting

Well-written and impressive systematic review, requiring only minor revisions. Well-structured article with clear tables and figures, and a very illustrative.conceptual model. Aside from the description of FBT in the introduction, the article is almost ready for publication.

Language: Except for a couple of sentences where a word is missing, the article uses clear and professional English

Rationale and Contribution Statement
In FBT, parents play a central role in tackling the eating disorder, making the treatment’s effectiveness reliant on them. Since we know that FBT doesn’t work for everyone, it is highly relevant to explore which caregiver factors influence the delivery and outcomes of FBT.

Introduction:
The initial description of FBT seems somewhat simplified and quite brief compared to the otherwise thorough nature of this systematic review. It is therefore recommended to briefly introduce the core concept of FBT – that parents are viewed as a resource and the key factor in overcoming the eating disorder, and that they are instructed and supported in taking responsibility for the young person’s renourishment..etc. Perhaps also include a brief overview of the three phases of FBT, as Phase 1 is later referenced in the manuscript.

The sentence ‘Motivation to activate caregiver participation is grounded in the therapeutic advice that without nutritional rehabilitation, their child may die’ seems exaggerated – both in relation to the literature referenced and in clinical practice, where parents are largely supported and motivated - not simply told that they must take responsibility for the young person’s renourishment or their child may die.

Experimental design

The section ‘Gaps in Literature’ is well-argued and convincing. It is correct that there is currently no protocol for recommending a specific version of FBT to families. Moreover, since FBT views caregivers as crucial resources in their child’s eating disorder treatment, it is important to consider parental capacity and potential barriers to participation when constructing a useful case formulation for providing treatment.

Research Objective
Clear purpose, and positive that there is a specific focus on practical applicability in clinical practice

Method
This review includes all available research on caregiver factors, providing a comprehensive overview of the factors clinicians and researchers should focus on in future clinical work
The review has been registered with PROSPERO. The search strategy is structured according to the PICO framework, and the review follows PRISMA guidelines. The search was con-ducted across six databases. The heterogeneity of the data precluded meta-analysis; therefore, the SWiM guidelines were used to synthesise and group the results using a systematic-narrative approach. Overall, a thorough and well-written description of the search strategy and methodology. An impressively detailed and well-conducted review!

Results
A good overview of the eight domains, which impressively summarise a wide range of diverse findings.

The sentence on line 387 seems to be missing a word “Examination of the relationship between caregiver eating behaviors an outcome including adolescent eating disorder symptoms and weight gain found that a positive correlation to weight gain, and a negative correlation to symptoms”

Care-FIT Conceptual Model of Caregiver Factors Inûuencing Treatment: It is a very useful model that can be used to assess suitability for FBT, as well as for supervision, etc.

Validity of the findings

Discussion
I normally like that discussion begins with a recap of the findings. That is, the review included 39 studies, and the recurring variables measured within the studies informed the generation of eight domains, which could then eventually be listed.

Next, it is fint to pinpoint, as the review does, that the central concepts of self-efficacy and expressed emotion appear to be used to refer to quite different phenomena.

The sentence on line 549 appears to be missing a word “It may be helpful to expand the use of MFT as an adjunct to FBT to support increased remission rates, in line with Funderud et al..s (2023) study that combined delivery”.

Clinical Implications and Future Research
Well-written section with valuable points for clinical practice, such as considering adding MFT for single parents.

Conclusion
Also, a clear and well-summarised conclusion. Overall, an impressive systematic review that encompasses findings from very different types of data and successfully generates domains that are organised into a model, which is useful and can guide both future research and clinical practice

Additional comments

No additional comments

---

## Round 0.2 · accepted · Accept

The authors did a good job in addressing all the suggested points.

·

Basic reporting

Fine

Experimental design

Fine

Validity of the findings

Fine

Additional comments

I thank the authors for a thorough revision of the manuscript. I have no further comments, but reccommend the acceptance of this manuscript.

·

Basic reporting

The authors have addressed all of my comments and incorporated the changes into the manuscript. I believe the article is very well done and ready for publication.

Experimental design

The authors have addressed all of my comments and incorporated the changes into the manuscript. I believe the article is very well done and ready for publication.

Validity of the findings

The authors have addressed all of my comments and incorporated the changes into the manuscript. I believe the article is very well done and ready for publication.

Additional comments

The authors have addressed all of my comments and incorporated the changes into the manuscript. I believe the article is very well done and ready for publication.